# Efficient Exact and Approximate Betweenness Centrality Computation for Temporal Graphs

## ABSTRACT

Betweenness centrality of a vertex in a graph evaluates how often the vertex occurs in the shortest paths. It is a widely used metric of vertex importance in graph analytics. While betweenness centrality on static graphs has been extensively investigated, many real-world graphs are time-varying and modeled as temporal graphs. Examples include social networks, telecommunication networks, and transportation networks, where a relationship between two vertices occurs at a specific time. Hence, in this paper, we target efficient methods for temporal betweenness centrality computation. We firstly propose an exact algorithm with the new notion of *time instance graph*, based on which, we derive a temporal dependency accumulation theory for iterative computation. To reduce the size of the *time instance graph* and improve the efficiency, we propose an additional optimization, which compresses the *time instance graph* with equivalent vertices and edges, and extends the dependency theory to the compressed graph. Since it is theoretically complex to compute temporal betweenness centrality, we further devise a probabilistically guaranteed, high-quality approximate method to handle massive temporal graphs. Extensive experimental results on real-world temporal networks demonstrate the superior performance of the proposed methods. In particular, our exact and approximate methods outperform the state-of-the-art methods by up to two and five orders of magnitude, respectively.

## CCS CONCEPTS

• **Information systems** → **Social networks**; • **Theory of computation** → *Graph algorithms analysis*.

## KEYWORDS

Temporal Graph, Temporal Path, Betweenness Centrality, Algorithm

**ACM Reference Format:**
Anonymous Author(s). 2024. Efficient Exact and Approximate Betweenness Centrality Computation for Temporal Graphs. In *Proceedings of the ACM Web Conference 2024 (WWW '24)*. ACM, New York, NY, USA, 14 pages. https://doi.org/XXXXXXX.XXXXXXX

**Relevance to Web Research:** Calculating or estimating the vertex importance of web-based temporal social networks is a critical and challenging task. Our work uses temporal betweenness centrality to

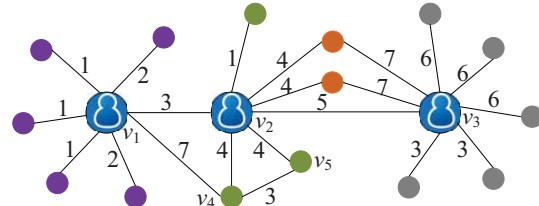

**Figure 1: A toy social network**

measure the importance of nodes, which is widely used in the study of influential node identification, information dissemination, and epidemic research. We provide exact and approximate algorithms for calculating vertex importance on temporal graphs, which make up for the shortcomings of previous algorithms in terms of time efficiency and spatial efficiency.

## 1 INTRODUCTION

Betweenness centrality (BC) measures the extent to which a vertex lies on the shortest paths between other vertices. Vertices with high betweenness centrality scores usually have a great influence on controlling the flow of information or dominating other entities' behavior. Currently, a vast majority of existing efforts are devoted to static graphs, in which the important temporal character is not considered. However, a substantial number of real-world graphs are time-varying and thus are usually modeled as temporal graphs, where edges are endowed with timestamps. Hence, in this paper, we focus on computing betweenness centrality for temporal graphs, which can be applied in epidemiology research [13], transmission of information [43], and brain disease research [23], to name but a few.

Example 1. *Figure 1 shows a toy social network, which is used for the study of epidemic. Vertices represent individuals and edges refer to the contacts. Considering that many possibly infective contacts between individuals are fleeting (e.g., people in the street or the marketplace) [19], each edge in Figure 1 is attached with a timestamp, denoting the fleeting contact. Naturally, epidemics can only flow along temporal paths, e.g., in Figure 1, diseases spread only from $v_1$ to $v_5$ via $v_2$, but not from $v_1$ to $v_5$ via $v_4$ because the link $(v_1, v_4)$ occurs after $(v_4, v_5)$. Temporal betweenness centrality measures the importance of a vertex based on the number of times that the vertex appears in an "optimal" temporal path between any pair of vertices [16]. The determination of the "optimal" temporal paths might depend either on time-related properties (e.g. earliest arrival time) or on the topological length. Vertices with high betweenness centrality scores (such as node $v_1$, $v_2$, $v_3$) are crucial for identifying super-spreaders to control the spreading of infectious diseases [13].*

When an extra dimension of time is added, BC computation on temporal networks is more challenging than on static graphs.

(i) *The various definitions of optimal temporal paths* [38] *on temporal graphs.* BC is a path-based metric. On static graphs, the straightforward definition of the shortest path is employed. On temporal graphs, however, various definitions of "optimal" temporal paths are used. Commonly used optimal temporal paths include the shortest and earliest temporal paths. They are defined over different criteria and applied in different applications. Thus, how to design a unifying approach that adapts to work with variants of "optimal" temporal paths is challenging. (ii) *Existing methods for static graphs are based on recursive dependency accumulation, which is invalidated on temporal graphs.* For temporal graphs, the optimal temporal path loses a critical property, i.e., *subpath optimality property, that is, the subpath of a shortest (optimal) path is still shortest (optimal)*. It is the key for computing BC in static graphs based on the pair dependency formulation proposed by Brandes [4]. Hence, existing theories and methods for static graphs cannot be directly applied to temporal graphs, new Temporal Betweenness Centrality (TBC) computation theory should be derived. (iii) *Exact algorithms have theoretically been proved to have high time complexity* [6]. The time complexity of computing exact TBC on a directed temporal graph is at least $O(n^3\mathcal{T}^2)$ ($n$ refers to the number of vertices and $\mathcal{T}$ refers to the total number of timestamps), counting earliest temporal paths is even #P-hard [6]. Such computational complexity is prohibitive for massive graphs.

To tackle the above challenges, we derive a new *recursive temporal dependency formulation* and present an exact algorithm ETBC (Exact Temporal Betweenness Centrality) to compute the TBC values. The theory is applied to the transformed *time instance graph* and is suitable for diverse optimal temporal paths. In order to reduce the scale of the *time instance graph* and thus speed up the computation, we further design a lossless compression method that compresses the *time instance graph* with equivalent vertices and edges, and then propose an optimized calculation theory and the optimized algorithm OTBC (Optimized Temporal Betweenness Centrality). Note that, though the state-of-the-art method [6] is also based on the idea of transformation, our proposed *time instance graph* is more dominant in three aspects. (i) The *time instance graph* is independent of the types of optimal temporal paths. That is to say, whether the optimal temporal path is shortest or earliest, the transformed *time instance graph* is the same. In contrast, the transformation of the predecessor graph proposed in the previous study [6] is strongly coupled with the types of optimal temporal paths, which means that different optimal temporal paths require constructing different predecessor graphs. (ii) The theory derived from the *time instance graph* adapts to work with shortest, earliest, and shortest earliest temporal paths, and can be easily extended to other types of optimal temporal paths. While the theory derived from the predecessor graph has limited adaptability, it cannot be applied to the earliest temporal path. (iii) The size of *time instance graph* is much smaller than that of the predecessor graph. Experiments show that on the infectious dataset, the predecessor graph is dozens of times larger than the *time instance graph*, and it consumes 90%+ memory.

In addition, to handle massive graphs, we propose an approximate method ATBC (Approximate Temporal Betweenness Centrality). The general idea of ATBC is to utilize the sampling method, which considers only a subset of vertices or temporal paths. To reach a certain accuracy with probabilistic guarantees and control the number of iterations, we utilize a tight error upper bound derived by Rademacher averages and integrate the optimized calculation theory with ATBC. In brief, our contributions are summarized as follows.

- We define a transformed *time instance graph*, which has characteristics of versatility and adaptivity. Then we derive a new recursive temporal dependency formulation and present an exact algorithm ETBC to compute TBC based on various optimal temporal paths.
- We devise a lossless compression method to reduce the scale of the *time instance graph*, and then propose an optimized calculation theory and the optimized algorithm OTBC to accelerate the TBC computation.
- To compute TBC in massive graphs, we devise a probabilistically guaranteed, high-quality approximate method ATBC, which utilizes a tight error upper bound derived by Rademacher averages and integrates the optimized calculation theory.
- Experiments conducted on 13 real temporal networks demonstrate the superior performance of the proposed methods. In particular, compared with the state-of-the-art exact and approximate methods, OTBC and ATBC are capable of up to two and five orders of magnitude speedup, respectively, and ATBC can scale to the large-scale graphs with 1.25M vertices and 20.26M edges.

Due to space restrictions, all proofs can be found in Appendix B.

## 2 RELATED WORK

**BC computation on static graphs.** Brandes' algorithm [4] and its improved versions, such as Brandes++ [10], BADIOS [32], are classic algorithms for computing exact BC values on static graphs. The main idea is to recursively compute the dependencies of a vertex $v$ on others. To alleviate the cost of BC computation, a set of works propose approximation methods [1, 5, 18, 27, 37] and distributed approaches [2, 9, 14, 31, 33, 33, 35, 36]. Ziyad et al. [1] propose a benchmark, called BeBeCA, for approximate BC computation. By the ways of sampling, the approximation algorithms are mainly divided into two categories, i.e., source sampling and node-pair sampling. Source sampling [5, 18] selects pivots with the highest degree, maximum or minimum distance strategies, then uses all pair dependencies that start from the selected pivots to approximate BC values. Node-pair sampling [3, 8, 25, 27, 28] either uses fixed-size sampling or progressive sampling to sample node-pairs at random uniformly, and then computes the pair dependencies of sampling node-pairs to approximate BC values. Another line of research defines variants of BC, e.g., $k$-BC [26], $\kappa$-path centrality [21], ego-BC [11], which are easier to compute. Recently, Graph Neural Network based inductive frameworks [12, 24] are presented, but they have no probabilistic guarantees on the output. In summary, either exact or approximate BC computation algorithms for static graphs are based on the *subpath optimality property*, which is invalid on temporal graphs.

**BC computation on temporal graphs.** Researches investigate queries or analysis tasks [17, 22, 29, 38, 40–42] on temporal

graphs [7, 15, 20, 39]. Here, we focus on related works on temporal paths and TBC calculation. Wang et al. [38] explored three types (i.e., earliest arrival path, latest departure path, shortest duration path) of route planning queries on timetable graphs. Wu et al. [40, 41] studied minimum temporal paths computation and temporal reachability queries. Buß et al. [6] theoretically investigate algorithmic aspects of temporal betweenness variants based on strict and non-strict shortest, the combination of shortest and earliest temporal paths. They construct a predecessor graph and adapt the approach of Brandes' algorithm to the predecessor graph. However, the proposed algorithm [6] cannot be applied to the earliest temporal path. Tsalouchidou et al. [34] consider the combination of the path length and time duration as an optimality criterion, and compute the exact TBC with static snapshots. These two methods have the defects of large memory consumption and low efficiency. ONBRA [30] is the first TBC approximation algorithm. It leverages empirical Bernstein bound to guarantee the accuracy, and estimates the TBC values based on the method proposed in [6].

## 3 PRELIMINARIES

DEFINITION 1. (**Temporal Graph**). *A directed un-weighted temporal graph is denoted by $G = (V, E, T)$, where (i) $V$ is a vertex set; (ii) $E$ is a directed temporal edge set, $e_i = (u, v, t_v) \in E$ represents a temporal edge from node $u$ to $v$, indicating an instantaneous event from $u$ to $v$ takes place at time $t_v \in T$; and (iii) $T$ is a finite temporal domain. $\forall w \in V$, $T_w = \{t_i \mid 1 \le i \le h\}$ is the set of distinct timestamps attached with the incoming edges of $w$ and $h$ is the number of distinct timestamps attached with the incoming edges of $w$.*

Let $N_{out}(u) = \{v \mid (u, v, t_v) \in E\}$ refer to the set of $u$'s out-neighbors; $N_{in}(u) = \{w \mid (w, u, t_u)\}$ refer to the set of $u$'s in-neighbors; and $N(u) = N_{in}(u) \cup N_{out}(u)$ be the set of $u$'s neighbors.

DEFINITION 2. (**Temporal Path**). *A temporal path $p = u \xrightarrow{t_1} w_1 \cdots \xrightarrow{t_{m-1}} w_{m-1} \xrightarrow{t_m} v$, is defined as a sequence of contacts with non-decreasing time from $u$ to $v$, where for $1 \le i < m$, $t_i \le t_{i+1}$. We refer to $S_p = t_1$ as the start time of $p$, and $E_p = t_m$ as the end time of $p$. Further, we refer to $d(p) = m$ ( for unweighted temporal graphs, $d(p)$ is the number of edges in $p$; for weighted temporal graphs, $d(p)$ is the sum of the weights of the edges in $p$) as the length of $p$.*

Based on the definition of the temporal graph and temporal path, we introduce a function $TP_G(s, u)$ on a temporal graph $G$ that returns all directed temporal paths $p$ from a vertex $s$ to another vertex $u$. Then, we define two common types of optimal temporal paths, namely, *shortest temporal path* and *earliest temporal path* below.

DEFINITION 3. (**Shortest Temporal Path**). *The set of shortest temporal paths from $s$ to $u$ on $G$, denoted as $STP(s, u)$, includes the paths $p$ with the minimal length $\min(d(p'))$ among all paths $p'$ returned by $TP_G(s, u)$.*

DEFINITION 4. (**Earliest Temporal Path**). *The set of earliest temporal path from $s$ to $u$ on $G$, denoted as $ETP(s, u)$, retrieves the paths $p$ with the earliest end time $\min(E_{p'})$ among all paths $p'$ returned by $TP_G(s, u)$.*

DEFINITION 5. (**TBC**). *Given a temporal graph $G = (V, E, T)$, the normalized temporal betweenness centrality value $TBC(v)$ of any vertex $v \in V$ is defined as:*

$$TBC(v) = \frac{1}{|V|(|V| - 1)} \sum_{\forall s \ne v \ne z \in V} \frac{\sigma_{sz}(v)}{\sigma_{sz}}$$

where $|V|$ is the number of vertices in $G$; $\sigma_{sz}$ denotes the number of optimal temporal paths from $s$ to $z$; and $\sigma_{sz}(v)$ denotes the number of optimal temporal paths from $s$ to $z$ that $v$ goes though. For every vertex $v$ in $G$, $TBC(v)$ is proportional to the sum of the fractions of optimal temporal paths that go through $v$. The more the temporal paths pass through $v$, the higher the $TBC(v)$ value.

Table 5 in Appendix A summarizes our notation.

## 4 EXACT TBC COMPUTATION

### 4.1 TBC Iterative Accumulation

Optimal temporal path counting on temporal graphs is much harder than that on static graphs, because the recursive dependency accumulation proposed by Brandes [4] does not work. The key foundation of the recursive dependency accumulation holds is that the subpaths of shortest paths are still the shortest. Unfortunately, it is not satisfied in temporal graphs, i.e., the subpaths of optimal temporal paths may not be optimal. Hence, new techniques specialized for temporal graphs urgently need to be developed. Inspired by [44], we want to transform the temporal graph into a static graph first, and then try to derive a new temporal dependency accumulation based on the transformed graph to efficiently compute TBC values. The transformed graph is named as *time instance graph*.

**Time instance graph.** Given a temporal graph $G = (V, E, T)$, the time instance graph of $G$, denoted by $G_t = (V_t, E_t)$, is defined as:

**Vertices in $V_t$.** Each vertex $v \in V$ is transformed into a set of vertices $S(v)$ in $V_t$, where $S(v) = \{(v, t_v) \mid t_v \in T_v\}$. Here, $T_v$ is the set of distinct time instances attached with the incoming edges of $v$. To distinguish vertex $v$ in original temporal graph, in the following, we call the vertex $(v, t_v)$ in the time instance graph *vertex instance* of $v$. Note that, for any edge $(u, v, t_v) \in E$, if the timestamps associated with the incoming edges of $u$ are all greater than $t_v$, i.e., $\forall t_u \in T_u$, $t_u > t_v$ holds, then $u$ is called a *source vertex*. Specially, if a vertex $v \in V$ has no incoming temporal edges, $v$ is also a *source vertex*. For each source vertex $v$, in the *time instance graph*, there is a vertex instance $(v, MIN)$ of $v$, $(v, MIN) \in S(v)$, where $MIN$ denotes the minimum timestamp.

**Edge in $E_t$.** For each edge $(u, v, t_v) \in E$, for each vertex instance $(u, t_u) \in S(u)$, we create an edge $\{(u, t_u), (v, t_v)\}$ from $(u, t_u)$ to $(v, t_v)$ if $t_u \le t_v$.

The graph transformation algorithm is outlined in Appendix C.1.

LEMMA 1. *The optimal temporal paths from any vertex $u$ to $v$ in $G$ can be exactly computed by traversing the paths from $u$ to all vertex instances of $v$ in $G$'s time instance graph $G_t$.*

Brandes [4] defines the dependency of $s \in V$ on a vertex $v$ as $\delta_{s.}(v) = \sum_{s \ne v \ne z \in V} \delta_{sz}(v) = \sum_{s \ne v \ne z \in V} \frac{\sigma_{sz}(v)}{\sigma_{sz}}$, i.e., the ratio of shortest paths from $s$ that $v$ lies on. In this paper, similarly, we introduce the temporal dependency below.

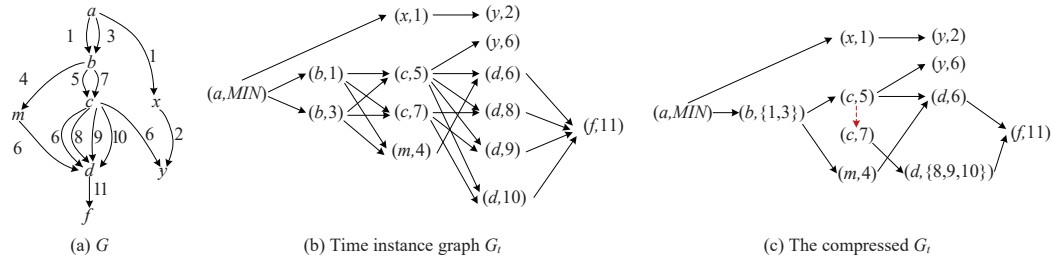

(a) $G$  (b) Time instance graph $G_t$  (c) The compressed $G_t$

**Figure 2: Example of a temporal graph, time instance graph, and the compressed time instance graph**

DEFINITION 6. *The temporal dependency of $s$ on a vertex instance $(v, t_v)$ is defined as $\delta_{s.}(v, t_v) = \sum_{s \neq v \neq z \in V} \delta_{sz}(v, t_v) = \sum_{s \neq v \neq z \in V} \frac{\sigma_{sz}(v, t_v)}{\sigma_{sz}}$, where $\sigma_{sz}(v, t_v)$ is the number of optimal temporal paths from $s$ to $z$ via $v$ at time $t_v$. $\sigma_{sz}$ is the number of optimal temporal paths from $s$ to $z$.*

LEMMA 2. *The temporal dependency $\delta_{s.}(v, t_v)$ of $s$ on $(v, t_v)$ obeys:*

$$\delta_{s.}(v, t_v) = \sum_{s \neq v \neq z \in V} \delta_{sz}(v, t_v) = \sum_{s \neq v \neq z \in V} \frac{\sigma_{sz}(v, t_v)}{\sigma_{sz}}$$
$$= \sum_{(v, t_v) \in P_s(w, t_w)} \left( \frac{\sigma_{s(v, t_v)} \cdot Flag(w, t_w)}{\sum_{t' \in T_w} \sigma_{s(w, t')} \cdot Flag(w, t')} \right.$$
$$\left. + \frac{\sigma_{s(v, t_v)}}{\sigma_{s(w, t_w)}} \cdot \delta_{s.}(w, t_w) \right),$$

*where $\sigma_{s(v, t_v)}$ is the number of local optimal paths from $s$ to the vertex instance $(v, t_v)$. Note that, we refer to the optimal path from $s$ to the vertex instance $(v, t_v)$, obtained by traversing the time instance graph $G_t$, as the "local optimal path" for the sake of distinction, because the local optimal path may not be the real optimal path from $s$ to $v$. $P_s(w, t_w)$ is the list of predecessors of a vertex instance $(w, t_w)$ on local optimal paths from $s$. As the local optimal paths from $s$ to $(w, t_w)$ computed in $G_t$ may not be the optimal temporal paths from $s$ to $w$, $Flag(w, t_w)$ is used to indicate whether the vertex instance $(w, t_w)$ is the end vertex of an optimal temporal path from $s$ to $w$. $Flag(w, t_w)$ is either 0 or 1, $Flag(w, t_w) = 1$ means that $(w, t_w)$ is the end vertex, otherwise $Flag(w, t_w) = 0$.*

Lemma 2 shows that the dependency of $s$ on the vertex instance $(v, t_v)$ can be computed using the dependencies of $s$ on the successors of $(v, t_v)$. Note that, not all the successors have contribution to the temporal dependency score. Only successors $(w, t_w)$ that are end vertices of optimal temporal paths from $s$ contribute to $\delta_{s.}(v, t_v)$. For instance, in Figure 2(b), $a \rightarrow (b, 1)$ (or $(b, 3)) \rightarrow (c, 5) \rightarrow (y, 6)$ is a local shortest path, and $a \rightarrow (x, 1) \rightarrow (y, 2)$ is the optimal shortest path from $a$ to $y$, which show that $(y, 2)$ is the end vertex of an optimal temporal path from $a$ to $y$, while $(y, 6)$ is not. Hence, $Flag(y, 6) = 0$ and $(y, 6)$ has no contribution to $\delta_{a.}(c, 5)$, $\delta_{a.}(b, 1)$ (or $\delta_{a.}(b, 3)$). The dependency scores of all vertex instances can be iteratively accumulated by traversing the time instance graph in non-increasing distances from $s$.

LEMMA 3. *The TBC value of a vertex $v$ in the temporal graph $G$ is the sum of temporal dependencies of all vertices in $G$ on $v'$s vertex instances.*

$$TBC(v) = \frac{1}{|V|(|V| - 1)} \sum_{s \in V} \sum_{t_v \in T_v} \delta_{s.}(v, t_v)$$

---

**Algorithm 1:** ETBC algorithm

**Input:** time instance graph $G_t = (V_t, E_t)$
**Output:** $TBC(v)$ for each vertex $v \in V$

1: **foreach** $s \in V$ **do**
2:      $d(s, (w, t_w)), d(s, w), Flag(w, t_w), P_s(w, t_w), \sigma_{s(w, t_w)}, Q$
     $\leftarrow$ SSSP$(G_t, s)$
3:      **while** $Q$ is not empty **do**
4:        dequeue $(w, t_w) \leftarrow Q$
5:        **foreach** $(v, t_v) \in P_s(w, t_w)$ **do**
6:          compute $\delta_{s(w, t_w)}(v, t_v)$, $\delta_{s.}(v, t_v) \leftarrow \delta_{s.}(v, t_v) +$
         $\delta_{s(w, t_w)}(v, t_v)$ by Lemma 2
7:      $TBC(w) \leftarrow \frac{1}{|V|(|V|-1)} (TBC(w) + \delta_{s.}(w, t_w))$ by Lemma 3

8: **return** $TBC(v)$ for each vertex $v \in V$

---

### 4.2 Exact Algorithm

Lemma 2 together with Lemma 3 allows us to iteratively compute the $TBC(v)$ by traversing the *time instance graph*. We propose an Exact TBC computation algorithm ETBC based on the definition of STP, while calculating TBC based on ETP is similar. The pseudo-code of ETBC is shown in Algorithm 1. ETBC consists of two phases. In the first phase, ETBC runs a single-source shortest path SSSP algorithm (e.g., breadth-first search (BFS) for un-weighted temporal graph; Dijkstra's algorithm for weighted temporal graph) from every vertex $s$ to compute the shortest paths to all the other vertex instances, and maintains (i) the local shortest path distance $d(s, (w, t_w))$ from $s$ to any vertex instance $(w, t_w)$; (ii) the shortest temporal path distance $d(s, w) = min_{t_w \in T_w} d(s, (w, t_w))$ from $s$ to $w$; (iii) the flag $Flag(w, t_w)$ that indicates whether $(w, t_w)$ is the end vertex of an optimal temporal path from $s$ to $w$; (iv) the list $P_s(w, t_w)$ of predecessors for $(w, t_w)$ on all local shortest paths from $s$; (v) the queue $Q$ that stores the vertex instances in nondecreasing shortest path distance from $s$; and (vi) the count $\sigma_{s(w, t_w)}$ of local shortest paths from $s$ to $(w, t_w)$ (line 2). Lemma 1 shows that traversing the *time instance graph* correctly finds all the shortest temporal paths from $s$ to any vertex $w$. In the second phase, ETBC iteratively dequeues the vertex instances from $Q$ and accumulates temporal dependencies by applying Lemma 2 (lines 3-6). At the end of each iteration, the temporal dependencies of $s$ on each other vertex instance $(z, t_z)$ are added to corresponding $TBC(z)$ according to Lemma 3 (line 7).

**Time Complexity.** The core part of ETBC is single-source shortest path computation, which can be determined in $O(|V_t| + |E_t|)$ for un-weighted temporal graphs or $O(|E_t| + |V_t| log |V_t|)$ for weighted

graphs. Therefore, the computational complexity for ETBC is $O(|V_t| (|E_t| + |V_t| log|V_t|))$ for weighted temporal graphs and $O(|V_t|(|V_t|+ |E_t|))$ for un-weighted temporal graphs, where $|V_t|$ and $|E_t|$ are the number of vertex instances and edges in $G_t$, respectively.

## 4.3 Optimization Techniques

We present compression techniques to reduce the number of vertex instances and edges in the *time instance graph*, and thus optimize TBC computation.

**Rule 1: Vertex instance compression.** We define equivalent vertex instances whose temporal dependencies are the same and compress these vertex instances to avoid extra computation.

**Equivalent vertex instances.** For the vertex instances $(w, t_1)$, $(w, t_2)$ of the same vertex $w$, $(w, t_1)$ and $(w, t_2)$ are equivalent if and only if their in-neighbors and out-neighbors are same, i.e., $N_{in}(w, t_1) = N_{in}(w, t_2)$, $N_{out}(w, t_1) = N_{out}(w, t_2)$. The equivalent vertex instances $(w, t_1)$, $(w, t_2)$ are compressed into a vertex instance $(w, \{t_1, t_2\})$. Let $Ident(w, t_w)$ denote the number of equivalent vertex instances. Initially, $Ident(w, t_w) = 1$ for all $(w, t_w) \in V_t$.

**Rule 2: Edge compression.** For the vertex instances $(w, t_1)$, $(w, t_2), \cdots, (w, t_n)$ of the same vertex $w$, if $N_{in}(w, t_1) = N_{in}(w, t_2)$ $\cdots = N_{in}(w, t_n)$, then do the following edge compression: We sort the vertex instances $(w, t_1)$, $(w, t_2), \cdots, (w, t_n)$ in nondecreasing timestamp, and then create *virtual edges* between adjacent vertex instances. At the same time, (i) For each in-neighbor $(u, t_u)$ in $N_{in}(w, t_i)(1 \leq i \leq n)$, we remove all the edges from $(u, t_u)$ to $(w, t_i)(1 \leq i \leq n)$ except edge $\{(u, t_u), (w, t_j)\}$ having $t_j = min\{t \mid t \geq t_u, t \in \{t_1, t_2, \cdots, t_n\}\}$; (ii) For the same out-neighbor $(v, t_v)$ in $N_{out}(w, t_i)(1 \leq i \leq n)$, we remove all the edges from $(w, t_i)(1 \leq i \leq n)$ to $(v, t_v)$ except edge $\{(w, t_i), (v, t_v)\}$ having $t_k = max\{t \mid t \leq t_v, t \in \{t_1, t_2, \cdots, t_n\}\}$.

Note that, the weights of *virtual edges* are 0. On one hand, *virtual edges* pass the value of $\sigma_{s(w, t_1)}$ to $(w, t_2), \cdots, (w, t_n)$; on the other hand, the temporal dependencies on successors are propagated to $(w, t_n)$, as well as $(w, t_{n-1}), \cdots, (w, t_1)$ along *virtual edges*. However, the temporal dependency on $(w, t_n)$ should not be propagated to $(w, t_1)$, $(w, t_2), \cdots, (w, t_{n-1})$ via *virtual edges*, because $(w, t_1)$, $(w, t_2), \cdots, (w, t_n)$ are the instances of the same vertex $w$.

EXAMPLE 2. *Take Figure 2 as an example. Figure 2(b) shows the time instance graph $G_t$ of the temporal graph $G$ depicted in Figure 2(a). The final compressed time instance graph is shown in Figure 2(c). First, $(b, 1)$, $(b, 3)$ are equivalent and are compressed into $(b, \{1, 3\})$, as $N(b, 1) = N(b, 3) = \{(a, MIN), (c, 5), (c, 7), (m, 4)\}$. Then, for $(c, 5)$ and $(c, 7)$, $N_{in}(c, 5) = N_{in}(c, 7) = \{(b, \{1, 3\})\}$, hence we do edge compression. A virtual edge (shown in red dotted in Figure 2(c)) $\{(c, 5), (c, 7)\}$ is created, $\{(b, \{1, 3\}), (c, 7)\}$, $\{(c, 5), (d, 8)\}$, $\{(c, 5), (d, 9)\}$, $\{(c, 5), (d, 10)\}$ are removed. In the sequel, $(d, 8)$, $(d, 9)$, $(d, 10)$ are identified as equivalent vertex instances, then they are compressed into $(d, \{8, 9, 10\})$, and $Ident(d, \{8, 9, 10\}) = 3$.*

LEMMA 4. *After compression, the temporal dependency $\delta_{s.}(v, t_v)$ of $s$ on $(v, t_v)$ is optimized as:*

$$\delta_{s.}(v, t_v) = \sum_{(v, t_v) \in P_s(w, t_w)} \left( \frac{\sigma_{s(v, t_v)} \cdot Flag(w, t_w)}{\sum_{t' \in T_w} \sigma_{s(w, t')} \cdot Flag(w, t')} \right.$$
$$\left. \cdot Ident(w, t_w) + \frac{\sigma_{s(v, t_v)} \cdot Ident(w, t_w)}{\sigma_{s(w, t_w)}} \cdot \delta_{s.}(w, t_w) \right),$$

where

$$\sigma_{s(w, t_w)} = \begin{cases} \sum_{\forall (v, t_v) \in P_s(w, t_w)} \sigma_{s(v, t_v)} \cdot Ident(w, t_w), & s \neq v \\ \sigma'_{s(w, t_w)} \cdot Ident(w, t_w), & otherwise \end{cases}$$

$\sigma'_{s(w, t_w)}$ *is the number of local optimal paths from $s$ to $(w, t_w)$ computed by traversing the compressed time instance graph.*

Based on $G_t$, we present OTBC algorithm, which optimizes the ETBC ($\delta_{s.}(v, t_v)$) computation in line 6) with Lemma 4. In addition, the calculation of optimal temporal paths from different sources and the summation of the temporal dependencies to compute the TBCs are independent, and thus they could be completely parallel (i.e., lines 1-7 of Algorithm 1 are paralleled computed) using OpenMP. OpenMP delegates a user-specified number of threads to compute the number of optimal temporal paths from different sources to other vertices in parallel. During the computation, different threads create their own local memory capacity, and they do not share any local data. At the end, different threads need to add the TBCs of vertex instances to the global TBCs, which are maintained by an array.

Due to limited space, an example that shows how $\delta_{s.}(v, t_v)$ is computed by Lemma 4 is detailed in Appendix D. The graph compression algorithm and theoretical complexity analyses are provided in Appendix C. The compression ratios, how to support various optimal paths, and how to handle graph updates are discussed in Appendix E.

## 5 APPROXIMATE TBC COMPUTATION

Today's temporal networks are massive, maintaining the TBC value on temporal graphs with millions of nodes and tens of millions of temporal edges is prohibitive. A high-quality approximate computation is more feasible. Hence, based on the compressed *time instance graph*, we further propose an approximate TBC computation method, called ATBC, which guarantees that all approximate TBC values are within an additive factor $\epsilon \in (0, 1)$ from the real values, with probability at least $1 - \delta$. i.e.,

$$Pr(\forall w \in V, |\widehat{TBC(w)} - TBC(w)| \leq \epsilon) \geq 1 - \delta.$$

ATBC is inspired by the idea of ABRA [28], which exploits *Rademacher Averages* to derive an upper bound to the maximum deviation of the approximate values from exact ones, and then uses progressive random sampling to iterative compute approximate TBC value that is an $(\epsilon, \delta)$-approximation. Fortunately, the upper bound derived by Rademacher Averages is also suitable for TBC estimation. Here we directly present the upper bound. While the in-depth proof and discussion are not elaborated. We refer the readers to the literature [28] for details.

Let the finite domain $\mathcal{R}$ be the pairs of different nodes, and $f_w : \mathcal{R} \rightarrow [0, 1]$ be the function, which is defined as $f_w(s, z) = \frac{\sigma_{sz}(w)}{\sigma_{sz}}$. $\mathcal{F} = \{f_w, w \in V\}$ is a family of functions from $\mathcal{R}$ to $[0, 1]$. Let $\mathcal{S}$ be a set of vertex pairs sampled from $\mathcal{R}$. For each sampled vertex pair $(u_i, v_i) \in \mathcal{S}$, $\mathbf{v}_w = (f_w(u_1, v_1), f_w(u_2, v_2), \cdots, f_w(u_{|\mathcal{S}|}, v_{|\mathcal{S}|}))$ is a vector, and $\mathcal{V}_S = \{\mathbf{v}_w, w \in V\}$. $\eta \in (0, 1)$ is a parameter. Then, the upper bound $\tilde{ub} = sup|TBC(w) - \widehat{TBC(w)}|$ to the maximum

deviation is:

$$\bar{ub} \leq \frac{\omega^*}{1-\alpha} + \frac{ln\frac{2}{\eta}}{2|\mathcal{S}|\alpha(1-\alpha)} + \sqrt{\frac{ln\frac{2}{\eta}}{2|\mathcal{S}|}}$$

$$\alpha = \frac{ln\frac{2}{\eta}}{ln\frac{2}{\eta} + \sqrt{(2|\mathcal{S}|\omega^* + ln\frac{2}{\eta})ln\frac{2}{\eta}}}$$

$$\omega^* = min_{c\in\mathbb{R}^+}\frac{1}{c}ln\sum_{\mathbf{v}\in\mathcal{V}_\mathcal{S}} exp(c^2||\mathbf{v}||^2/2|\mathcal{S}|^2) \quad (1)$$

Based on the derived upper bound, ATBC is developed. In summary, ATBC is an iterative algorithm, which samples vertex pairs and iteratively computes $\widetilde{TBC(w)} = ||\mathbf{v}_w||/|\mathcal{S}|$ ($\forall w \in V$) by the optimized calculation theory until a $(\epsilon, \delta)$-approximation is achieved. In each iteration, ATBC first sample a set of original vertex pairs. For each pair $(u, v)$, ATBC performs the single-source shortest path algorithm from $u$ to all $v$'s vertex instances $(v, t_v)$ on the compressed temporal graph. For all the traversed vertex instances $(z, t_z)$, ATBC computes predecessors, the shortest temporal path distance, the flag $Flag(v, t_v)$ that indicates whether $(v, t_v)$ is the end vertex of an optimal temporal path from $u$ to $v$, and the count $\sigma_{u(z,t_z)}$ of local shortest paths by Eq. 1. If all the vertex instances of $v$ are traversed, ATBC computes $\sigma_{uv}$, starts backtracking $(w, t_w)$ from all vertex instances $(v, t_v)$ of $v$ having $Flag(v, t_v) = 1$ to $u$ along the computed shortest paths, computes $\sigma_{uv}(w) = \sum_{(w,t_w)\in S(w),Flag(v,t_v)=1} \sigma_{u(v,t_v)}(w, t_w)$, $f_w(u, v) = \frac{\sigma_{uv}(w)}{\sigma_{uv}}$, and updates vector $\mathbf{v}_w$. Thereafter, ATBC checks whether $(\epsilon, \delta)$-approximation is satisfied (i.e., the stopping condition $\bar{ub} \leq \epsilon$ holds). The pseudocode of ATBC and complexity analysis are provided in Appendix C.

# 6 EXPERIMENTAL EVALUATION

**Datasets**. We employ 13 real temporal graphs. The statistics are summarized in columns 2-4 of Table 1, where $|V|$, $|E|$ and $|T|$ are the number of vertices, the number of edges, and the number of distinct timestamps, respectively. The datasets from highschool-2011 to infectious are from SocioPatterns[1]; emails, wikivoyage-it, and wikiedits-se are from Konect[2]; mathoverflow, superuser, wikitalk are from the SNAP[3]. For every dataset, we performed ten runs for each combination of parameters and randomly reported the results for a run as the variance between the different runs was essentially insignificant.

**Methods**. We compare the proposed exact algorithm ETBC, optimized algorithm OTBC, and approximate algorithm ATBC with the following exact approaches: (i) sliding temporal window based method TBC [34][4], which considers the combination of the path length and the time duration as an optimality criterion; (ii) state-of-the-art exact TBC computation method, called KDD-TBC [6][5]; and (iii) state-of-the-art approximate TBC computation method ONBRA [30][6].

---

[1]SocioPatterns is available at http://www.sociopatterns.org/datasets/.
[2]Konect is available at http://konect.cc/
[3]SNAP is available at https://snap.stanford.edu/data/.
[4]Code of TBC is available at https://goo.gl/PAAJvp.
[5]Code of KDD-TBC is available at http://fpt.akt.tu-berlin.de/software/temporal_betweenness/.
[6]Code of ONBRA is available at https://github.com/iliesarpe/ONBRA.

All methods were implemented in C++, and run on a Ubuntu machine of 128G memory and two Intel(R) Xeon(R) E5-2640 2.40GHz CPU. Particularly, in each experiment, we use parallel techniques based on OpenMP to speed up centrality computation, and set the number of threads to 24 to achieve good performance. Note that we use bold values in the tables to highlight the best results, "—" to indicate that an algorithm cannot return the result within 24 hours, and "OOM" to represent that a method runs out of memory.

## 6.1 Statistics of Time Instance Graph

Table 1 presents the scale of the transformed and compressed *time instance graphs*, as well as the compression ratios. Specifically, compression ratios includes vertex compression ratio *vc-ratio* and edge compression ratio *ec-ratio*, which are defined as $vc\text{-}ratio = 1 - \frac{|V_t'|}{|V_t|}$ and $ec\text{-}ratio = 1 - \frac{|E_t'|}{|E_t|}$. The larger the compression ratio is, the more effective the compression strategies are. First, it is observed that the number of vertices in the transformed *time instance graphs* is from 4 to 505 times larger than that in the original temporal graphs, and the number of edges in the transformed *time instance graphs* is from 6 to 651 times larger. Second, as shown in Table 1, *vc-ratio* and *ec-ratio* are in the range of (0.1, 0.8) and (0.2, 1), respectively, which shows the effectiveness of the compression strategies and the estimation method proposed in Section 4.3.

## 6.2 Performance of Exact Methods

**TBC Performance based on STP**. Following KDD-TBC [6], we distinguish strict (-Str) and non-strict (-Nstr) temporal paths by whether the timestamps of consecutive edges in a temporal path are strictly ascending or non-strictly ascending. Note that, on wikivoyage-it, wikiedits-se, mathoverflow, superuser, and wikitalk, KDD-TBC runs out of memory with a single thread. On infectious, the reported time is obtained using a single thread, because KDD-TBC requires 120.9 GB of RAM peak in the single-threaded environment, and thus cannot run in the multi-threaded environment. As shown in Table 2, the first observation is that the performance of TBC computation based on non-strict and strict temporal paths are very similar, ETBC and OTBC are clearly faster than comparable approaches over different datasets. For example, the computation cost of OTBC-Str is between 9 and 525 times less than that of KDD-TBC-Str; the running time of ETBC-Str is between 4 and 310 times less than that of KDD-TBC-Str. The reason is that our defined *time instance graph* preserves all temporal reachabilities, based on which, temporal dependency formulations proposed by Lemmas 2 and 4, and multi-thread technique enables ETBC and OTBC iteratively accumulate the temporal dependencies for fast computing TBC values. The second observation is that OTBC outperforms ETBC on all datasets due to the effective compression strategies as analyzed in Section 6.1 and the optimized temporal dependency enabled by Lemma 4. The third observation is that KDD-TBC runs out of memory on massive datasets, TBC cannot return the result within 24 hours on most of the datasets. In contrast, on mathoverflow, OTBC can complete the TBC computation in 26min. This is because TBC needs to set a large window size when the timestamps of the dataset have a great dispersion, resulting in lots of time to copy the graph data in the static window. KDD-TBC needs to construct predecessor graphs.

**Table 1: Statistics of the datasets, transformed and compressed time instance graphs, and the compression ratio**

| Datasets | Temporal graph | | | | Time instance graph | | | Compressed time instance graph | | | | Compression | |
|---|---|---|---|---|---|---|---|---|---|---|---|---|---|
| | $|V|$ | $|E|$ | $|T|$ | $|V_t|$ | $|E_t|$ | Size (MB) | Time(s) | $|V'_t|$ | $|E'_t|$ | Size (MB) | Time (s) | vc-ratio | ec-ratio |
| highschool-2011 | 126 | 28,560 | 5,609 | 25,665 | 1,904,005 | 14.82 | 0.3 | 8,068 | 232,712 | 1.9 | 0.37 | 0.686 | 0.878 |
| highschool-2012 | 180 | 45,047 | 11,272 | 41,042 | 6,189,685 | 47.69 | 0.8 | 12,874 | 462,442 | 3.73 | 1.32 | 0.686 | 0.925 |
| highschool-2013 | 327 | 188,508 | 7,375 | 165,715 | 39,991,015 | 307 | 19.54 | 62,924 | 4,528,189 | 35.42 | 11.83 | 0.620 | 0.887 |
| hypertext | 113 | 20,818 | 5,246 | 18,603 | 1,684,899 | 13.06 | 0.86 | 9,045 | 399,944 | 3.15 | 1.17 | 0.514 | 0.763 |
| hospital-ward | 75 | 32,424 | 9,453 | 27,875 | 10,161,230 | 77.84 | 1.36 | 14,141 | 1,842,198 | 14.21 | 4.15 | 0.493 | 0.819 |
| primaryschool | 242 | 125,773 | 3,100 | 98,836 | 21,985,732 | 168.86 | 8.86 | 69,768 | 8,345,334 | 64.31 | 20.75 | 0.294 | 0.620 |
| infectious | 10,972 | 415,912 | 76,944 | 339,836 | 7,343,768 | 59.91 | 2.09 | 209,145 | 3,094,542 | 25.72 | 4.74 | 0.385 | 0.579 |
| emails | 167 | 82,927 | 57,791 | 82,543 | 54,108,283 | 413.75 | 8.86 | 73,089 | 40,416,483 | 308.94 | 20.75 | 0.115 | 0.253 |
| wikivoyage-it | 31,501 | 419,474 | 404,760 | 417,258 | 20,709,922 | 162.77 | 0.57 | 96,059 | 571,135 | 6.31 | 4.82 | 0.770 | 0.972 |
| wikiedits-se | 18,055 | 261,169 | 258,625 | 261,188 | 1,861,159 | 17.18 | 3.01 | 132,318 | 543,915 | 5.65 | 11.32 | 0.493 | 0.708 |
| mathoverflow | 24,818 | 506,550 | 389,952 | 398,563 | 95,730,373 | 734.92 | 16.08 | 325,399 | 56,302,782 | 432.34 | 31.07 | 0.184 | 0.412 |
| superuser | 194,085 | 1,443,339 | 1,437,199 | 1,196,237 | 240,817,065 | 1850.98 | 52.68 | 1,053,956 | 148,487,326 | 1141.79 | 113.61 | 0.119 | 0.383 |
| wikitalk | 1,140,149 | 7,833,140 | 7,375,042 | 6,143,193 | 2,013,180,452 | 15429.65 | 661.23 | 4,766,098 | 1,228,161,722 | 9423.41 | 3078.4 | 0.224 | 0.390 |

**Table 2: TBC computation time based on STP (in seconds)**

| Datasets | ETBC-Str | OTBC-Str | KDD-TBC-Str | TBC-Str | ETBC-Nstr | OTBC-Nstr | KDD-TBC-Nstr | TBC-Nstr |
|---|---|---|---|---|---|---|---|---|
| highschool-2011 | 0.21 | **0.07** | 0.84 | 4,397.62 | 0.23 | **0.07** | 0.86 | 4,407.68 |
| highschool-2012 | 0.64 | **0.17** | 4.15 | 35,421.35 | 0.66 | **0.17** | 4.2 | 35,633.62 |
| highschool-2013 | 5.48 | **1.61** | 40.59 | — | 5.52 | **1.62** | 40.63 | — |
| hypertext | 0.38 | **0.16** | 1.55 | 4,460.36 | 0.38 | **0.18** | 1.54 | 4,473.61 |
| hospital-ward | 0.82 | **0.25** | 6.72 | 4,018.29 | 0.84 | **0.25** | 6.83 | 4,038.44 |
| primaryschool | 3.15 | **2.12** | 20.77 | 8,067.85 | 3.58 | **2.13** | 20.93 | 8,235.78 |
| infectious | 5.09 | **3.01** | 1,583.62 | — | 5.12 | **3.08** | 1,590.58 | — |
| emails | 10.05 | **9.55** | 109.44 | — | 11.05 | **10.23** | 109.53 | — |
| wikivoyage-it | 31.42 | **7.01** | OOM | — | 32.63 | **7.46** | OOM | — |
| wikiedits-se | 5.37 | **2.53** | OOM | — | 6.89 | **2.56** | OOM | — |
| mathoverflow | 1,709.55 | **1,517.13** | OOM | — | 1,717.96 | **1,522.64** | OOM | — |
| superuser | 30,926 | **27,030.7** | OOM | — | 31,100.5 | **27,069.8** | OOM | — |

**Table 3: TBC computation time based on ETP, and the combinations of STP and ETP (STP&ETP) (in seconds)**

| Datasets | ETP | | STP&ETP | | |
|---|---|---|---|---|---|
| | ETBC | OTBC | ETBC | OTBC | KDD-TBC |
| highschool-2011 | 12.08 | **1.24** | 0.23 | **0.08** | 0.89 |
| highschool-2012 | 732.07 | **6.81** | 0.69 | **0.17** | 4.42 |
| highschool-2013 | OOM | **134.39** | 5.92 | **1.62** | 42.61 |
| hypertext | 345.77 | **17.43** | 0.38 | **0.18** | 1.72 |
| hospital-ward | 244.54 | **28.93** | 0.91 | **0.25** | 7.68 |
| primaryschool | 1,582.80 | **358.56** | 3.61 | **2.21** | 21.78 |
| infectious | OOM | **3,375.48** | 5.75 | **3.36** | 1,597.59 |
| emails | OOM | OOM | 11.77 | **10.61** | 104.59 |
| wikivoyage-it | 119.35 | **11.13** | 36.71 | **7.47** | OOM |
| wikiedits-se | 8.24 | **4.71** | 7.02 | **2.75** | OOM |
| mathoverflow | OOM | — | 1,743.09 | **1,538.79** | OOM |
| superuser | OOM | OOM | 31,173.3 | **27,126.5** | OOM |

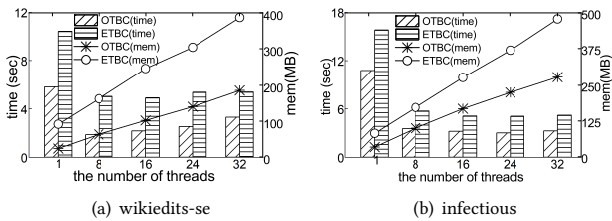

(a) wikiedits-se            (b) infectious

**Figure 3: Performance versus the number of threads**

**TBC Performance based on ETP and the combinations of STP and ETP.** KDD-TBC cannot be applied to ETP, and thus it is excluded for the case of ETP. As shown in Table 3, first, it is seen that the performance based on STP, the combinations of STP and ETP, is comparable; while ETP takes longer computation time than STP. Second, it is observed that, the improvement of OTBC on STP in Table 2 is not as significant as that on ETP in Table 3. This is because, either ETP or STP finds all the local optimal temporal paths by traversal. For STP, the first time the vertex instance $(v, t_v)$ is visited during the traversal, all the shortest paths from the source vertex to $(v, t_v)$ are found, other paths to $(v, t_v)$ are pruned and would not be extended. While this does not hold for ETP. During traversal, every time $(v, t_v)$ is visited, the path to $(v, t_v)$ is the ETP and should be extended, thus paths will be repeatedly traversed.

Compared to STP, OTBC acceleration of this part is amplified, and hence the improvement of OTBC on ETP is more significant. Last, we observe that OTBC achieves the highest efficiency, with ETBC in the second place, and then KDD-TBC.

**Effect of the number of threads.** We vary the number of threads from 1 to 32, and report the performance of ETBC and OTBC on wikiedits-se and infectious in Figure 3. As expected, memory usage linearly increases with the growth of the number of threads. In addition, it is observed that, the computation cost first sharply drops and then increases or stays stable when the number of threads (denoted by #thread) grows. The reason is that with more threads, there is more parallelism. But when the number of threads increases to a certain number, the thread overhead dominates the overall computing cost, and the costs of thread switching and data consistency guarantee increase. As tested, the optimal number of threads is decided by the scale of the input graph. For the small-scale or middle-scale datasets (such as wikivoyage-it, wikiedits-se),

Table 4: Efficiency and accuracy of ATBC (in seconds, $\delta = 0.1$)

| Datasets | $\epsilon$ | ATBC | Speedup w.r.t. | | OTBC | Sample size | Absolute Error($\times 10^{-5}$) | | | |
| | | | KDD-TBC | ONBRA | | | max | min | avg | stddev |
|---|---|---|---|---|---|---|---|---|---|---|
| infectious $|V'_t| = 215,006$ $|E'_t| = 3,207,294$ | 0.005 | 0.431 | 4,082 | 3,745 | 5.984 | 66,312 | 6.380 | 0 | 0.128 | 0.355 |
| | 0.010 | 0.179 | 9,835 | 4,336 | 15.816 | 17,657 | 11.084 | 0 | 0.171 | 0.705 |
| | 0.015 | 0.135 | 13,011 | 3,889 | 21.296 | 8,245 | 22.559 | 0 | 0.211 | 1.113 |
| | 0.020 | 0.104 | 16,909 | 3,684 | 27.942 | 5,034 | 57.514 | 0 | 0.246 | 1.701 |
| | 0.025 | 0.071 | 24,636 | 3,921 | 41.394 | 3,239 | 30.642 | 0 | 0.185 | 1.372 |
| | 0.030 | 0.062 | 28,447 | 3,783 | 47.548 | 2,414 | 80.907 | 0 | 0.221 | 2.074 |
| wikivoyage-it $|V'_t| = 124,371$ $|E'_t| = 658,702$ | 0.005 | 6.586 | | | 0.064 | 73,753 | 4.483 | 0 | 0.003 | 0.073 |
| | 0.010 | 3.496 | | | 1.005 | 18,718 | 22.343 | 0 | 0.005 | 0.178 |
| | 0.015 | 2.167 | OOM | OOM | 2.235 | 9,005 | 13.492 | 0 | 0.005 | 0.190 |
| | 0.020 | 1.439 | | | 3.871 | 5,172 | 44.946 | 0 | 0.008 | 0.380 |
| | 0.025 | 0.741 | | | 8.460 | 3,338 | 26.536 | 0 | 0.005 | 0.224 |
| | 0.030 | 0.553 | | | 11.676 | 2,339 | 43.699 | 0 | 0.005 | 0.306 |
| wikiedits-se $|V'_t| = 148,788$ $|E'_t| = 669,513$ | 0.010 | 2.325 | | >35,542 | 0.088 | 17,657 | 7.190 | 0 | 0.009 | 0.158 |
| | 0.015 | 1.196 | | >69,067 | 1.115 | 8,051 | 18.090 | 0 | 0.012 | 0.286 |
| | 0.020 | 0.812 | OOM | >101,781 | 2.116 | 4,929 | 36.926 | 0 | 0.013 | 0.420 |
| | 0.025 | 0.797 | | >103,623 | 2.174 | 3,215 | 31.054 | 0 | 0.020 | 0.568 |
| | 0.030 | 0.425 | | >194,263 | 4.953 | 2,332 | 42.662 | 0 | 0.014 | 0.511 |
| mathoverflow $|V'_t| = 342,044$ $|E'_t| = 56,386,288$ | 0.010 | 1,291.45 | | 27.09 | 0.175 | 35,723 | 82.493 | 0 | 0.581 | 2.654 |
| | 0.015 | 697.164 | | 34.23 | 1.176 | 16,108 | 155.582 | 0 | 0.867 | 4.205 |
| | 0.020 | 458.66 | OOM | 33.83 | 2.308 | 10,035 | 185.204 | 0 | 1.022 | 4.939 |
| | 0.025 | 285.127 | | 49.87 | 4.321 | 6,111 | 236.944 | 0 | 1.264 | 6.168 |
| | 0.030 | 218.605 | | 37.72 | 5.940 | 4,429 | 252.458 | 0 | 1.445 | 7.323 |
| superuser $|V'_t| = 841,242$ $|E'_t| = 181,514,409$ | 0.005 | 12,351.8 | | >5 | 2.188 | 119,378 | 98.310 | 0 | 0.092 | 0.605 |
| | 0.010 | 4,500.52 | | >17 | 5.006 | 31,968 | 77.100 | 0 | 0.153 | 0.920 |
| | 0.015 | 1,996.57 | | >39 | 12.539 | 14,284 | 144.020 | 0 | 0.195 | 1.345 |
| | 0.020 | 1,357.91 | OOM | >58 | 18.906 | 8,787 | 224.380 | 0 | 0.228 | 1.788 |
| | 0.025 | 817.101 | | >97 | 32.081 | 5,435 | 217.150 | 0 | 0.267 | 2.452 |
| | 0.030 | 624.973 | | >127 | 42.251 | 4,127 | 347.330 | 0 | 0.282 | 2.611 |
| wikitalk $|V'_t| = 5,855,430$ $|E'_t| = 1,231,309,541$ | 0.005 | 45,695.30 | | | >14 | 99,251 | — | — | — | — |
| | 0.010 | 26,779.00 | | | >22 | 24,958 | — | — | — | — |
| | 0.015 | 14,477.50 | OOM | OOM | >41 | 11,035 | — | — | — | — |
| | 0.020 | 8,326.55 | | | >72 | 6,855 | — | — | — | — |
| | 0.025 | 5,930.35 | | | >101 | 4,177 | — | — | — | — |
| | 0.030 | 3,227.01 | | | >187 | 3,020 | — | — | — | — |

#thread =8 or 16 is sufficient to enable the best performance. For massive-scale datasets (such as infectious, mathoverflow), #thread =24 or 32 is sufficient to enable the best performance.

## 6.3 Performance of ATBC

Table 4 plots the experimental results. We do not report the results on small-scale graphs because in these datasets, OTBC performs better than ATBC. The results for $\epsilon = 0.005$ are missing for mathoverflow and wikiedits-se because ATBC is little slower than OTBC. For ONBRA, the sample size $l$ is user-specified, we set $l$ properly such that the theoretical accuracy parameter $\epsilon$ computed by ONBRA is similar to ATBC. From Table 4, first it is observed that, KDD-TBC runs out of memory on most of datasets. ONBRA runs out of memory on wikivoyage-it and wikitalk, and does not finish within 23 hours on wikiedits-se and superuser when $\epsilon \approx 0.03$. ATBC is from several times to hundreds of times faster than OTBC; and it outperforms KDD-TBC and ONBRA by up to 28,447x and 194,263x, respectively. This is because, as shown in column 7 of Table 4, the sample size of ATBC is far less than the number of vertices in the time instance graphs, and ATBC integrates the optimized calculation theory, resulting in less computation cost. The different speedups for different $\epsilon$ are due to the different reduction in the sample size. Second, it is seen that, OTBC does not finish on wikitalk, but ATBC could complete the computation in 2 hours when $\epsilon \geq 0.025$. Third,

it is observed that, the running time of ATBC decreases quadratically with the growth of $\epsilon$. The reason is that, the larger the value of $\epsilon$ is, the smaller the sample size is, leading to less running time. Last, as depicted in columns 8-11 of Table 4, it is seen that, the minimum error is always 0; the maximum absolute error is at least an order of magnitude smaller than $\epsilon$ (not just with probability $> 1 - \delta$) with a very small standard deviation. These findings indicate that ATBC is an efficient, scalable, and an accurate (more that what is guaranteed by theoretical analysis) approach.

## 7 CONCLUSIONS

In this paper, we develop a new temporal dependency accumulation theory and design ETBC for efficiently computing exact TBC values. To improve the efficiency, we further propose optimized and approximate techniques. Extensive experiments with 13 real datasets demonstrate the efficiency and accuracy of our proposed approaches. In the future, we plan to extend our proposed methods to the distributed pregel-like systems. Another promising direction is to investigate more efficient approximate methods (including sampling method design, more stronger stopping condition derivation, and ego-temporal betweenness centrality computation).

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

 

# A  NOTATION

Table 5 shows our commonly used symbols and their definitions.

### Table 5: Symbols and description

| Notation | Description |
|---|---|
| $G, G_t$ | a temporal graph and the time instance graph |
| $T_w$ | the set of distinct timestamps attached with the incoming edges of vertex $w$ |
| $(v, t_v)$ | a vertex instance of vertex $v$ |
| $N_{in}(u), N_{out}(u), N(u)$ | the set of in-neighbors, out-neighbors and neighbors of vertex $u$ |
| $p = u \xrightarrow{t_1} w_1 \cdots \xrightarrow{t_m} v$ | a temporal path from $u$ to $v$ |
| $S_p, E_p, d(p)$ | the start time, the end time, and the length of path $p$ |
| $STP(s, u)$ | the set of shortest temporal paths from $s$ to $u$ |
| $ETP(s, u)$ | the set of earliest temporal paths from $s$ to $u$ |
| $TBC(v)$ | the normalized temporal betweenness centrality value of $v$ |
| $\widehat{TBC(v)}$ | the normalized approximate temporal betweenness centrality value of $v$ |
| $S(v) = \{(v, t_v) \mid t_v \in T_v\}$ | the set of all vertex instances of $v$ |
| $\sigma_{sz}(v, t)$ | the number of optimal temporal paths from $s$ to $z$ via $v$ at time $t$ |
| $\sigma_{sz}(v)$ | the number of optimal temporal paths from $s$ to $z$ via $v$ |
| $\sigma_{sz}$ | the number of optimal temporal paths from $s$ to $z$ |
| $\sigma_{s(v, t_v)}$ | the number of local optimal temporal paths from $s$ to the vertex instance $(v, t_v)$ (see Lemma 2) |
| $\delta_{s \cdot}(v, t_v)$ | the temporal dependency of $s$ on a vertex instance $(v, t_v)$ (see Definition 6) |
| $Flag(w, t_w)$ | a flag indicates whether $(w, t_w)$ is the end of the optimal temporal path from a source vertex $s$ to $w$ |
| $P_s(w, t_w)$ | the list of predecessors of the vertex instance $(w, t_w)$ on local optimal paths from $s$ |
| $d(s, (w, t_w))$ | the local optimal temporal path from $s$ to $(w, t_w)$ |
| $d(s, w) = \min_{t_w \in T_w} d(s, (w, t_w))$ | the optimal temporal path from $s$ to $w$ |
| $Ident(w, t_w)$ | the number of equivalent vertex instances of $(w, t_w)$ |

# B  OMITTED PROOFS

## PROOF OF LEMMA 1.

Proof. To guarantee the exactness of any optimal temporal path from $u$ to $v$, the core is that function $TP_G(u, v)$ finds all directed temporal paths $p$ from $u$ to $v$. Obviously, $TP_G(u, v) = \cup_{\forall (v, t_v) \in S(v)} P_{G_t}((u), (v, t_v))$. Here function $P_{G_t}((u), (v, t_v))$ returns the set of paths from $u$ to $(v, t_v)$ in $G_t$, where all the vertex instances of $u$ are reverted to the original vertex $u$, and the edges from different vertex instances of $u$ to another vertex instance $(w, t_w)$ are reduced to an edge from $u$ to $(w, t_w)$. Hence, the optimal temporal path from any vertex $u$ to $v$ in $G$ can be exactly computed. □

## PROOF OF LEMMA 2.

Proof.
$$\delta_{s\cdot}(v, t_v) = \sum_{s \neq v \neq z \in V} \delta_{sz}(v, t_v)$$
$$= \sum_{s \neq v \neq z} \sum_{(v, t_v) \in P_s(w, t_w)} \delta_{sz}((v, t_v), \{(v, t_v), (w, t_w)\})$$
$$= \sum_{((v, t_v) \in P_s(w, t_w)} \sum_{z \in V} \frac{\sigma_{sz}((v, t_v), \{(v, t_v), (w, t_w)\})}{\sigma_{sz}}$$

where $\delta_{sz}((v, t_v), \{(v, t_v), (w, t_w)\})$ is the temporal pair dependency that includes the edge $\{(v, t_v), (w, t_w)\}$. $\sigma_{sz}((v, t_v), \{(v, t_v), (w, t_w)\})$ is the number of optimal temporal paths from $s$ to $z$ via $v$ at time $t_v$ and the edge $(v, w, t_v)$. There are two cases:

(i) $z = w$. Then of the $\sigma_{sw}$ optimal temporal paths from $s$ to $w$, if $(w, t_w)$ is one of the end vertices of optimal temporal paths from $s$ to $w$, only $\sigma_{s(v, t_v)}$ first goes from $s$ to $(v, t_v)$, and then reach $(w, t_w)$ via edge $\{(v, t_v), (w, t_w)\}$, hence:

$$\delta_{sz}((v, t_v), \{(v, t_v), (w, t_w)\}) = \frac{\sigma_{s(v, t_v)} \cdot Flag(w, t_w)}{\sum_{t' \in T_w} \sigma_{s(w, t')} \cdot Flag(w, t')}$$

(ii) $z \neq w$. Then of the $\sigma_{sz}$ optimal temporal paths from $s$ to $z$, $\sigma_{s(v, t_v)}$ many first go from $s$ to $(v, t_v)$ and then use $\{(v, t_v), (w, t_w)\}$. Consequently, $\frac{\sigma_{s(v, t_v)}}{\sigma_{s(w, t_w)}} \cdot \sigma_{sz}(w, t_w)$ optimal temporal paths from $s$ to some $z \neq w$ contain $(v, t_v)$ and $\{(v, t_v), (w, t_w)\}$. Hence:

$$\delta_{sz}((v, t_v), \{(v, t_v), (w, t_w)\}) = \frac{\sigma_{s(v, t_v)}}{\sigma_{s(w, t_w)}} \cdot \frac{\sigma_{sz}(w, t_w)}{\sigma_{sz}}$$

Considering both cases (i) and (ii), we have:

$$\sum_{(v, t_v) \in P_s(w, t_w)} \sum_{z \in V} \frac{\sigma_{sz}((v, t_v), \{(v, t_v), (w, t_w)\})}{\sigma_{sz}}$$
$$= \sum_{(v, t_v) \in P_s(w, t_w)} \Big( \frac{\sigma_{s(v, t_v)} \cdot Flag(w, t_w)}{\sum_{t' \in T_w} \sigma_{s(w, t')} \cdot Flag(w, t')}$$
$$+ \frac{\sigma_{s(v, t_v)}}{\sigma_{s(w, t_w)}} \cdot \delta_{s\cdot}(w, t_w) \Big)$$

The proof completes. □

## PROOF OF LEMMA 3.

Proof. According to Definition 5 and Definition 6,
$$TBC(v) = \frac{1}{|V|(|V| - 1)} \sum_{\forall s \neq v \neq z \in V} \frac{\sigma_{sz}(v)}{\sigma_{sz}}$$
$$= \frac{1}{|V|(|V| - 1)} \sum_{s \in V} \delta_{s\cdot}(v)$$
$$= \frac{1}{|V|(|V| - 1)} \sum_{s \in V} \sum_{t_v \in T_v} \delta_{s\cdot}(v, t_v)$$

The proof completes. □

## PROOF OF LEMMA 4.

Proof. When traversing the compressed $G_t$, the edge $\{(v, t_v), (w, t_w)\}$ leads to $Ident(w, t_w)$ paths, and thus,

$$\sigma_{s(w, t_w)} = \sum_{\forall (v, t_v) \in P_s(w, t_w), s \neq v} \sigma_{s(v, t_v)} \cdot Ident(w, t_w)$$

otherwise we multiply the number of local optimal paths from $s$ to $(w, t_w)$ by $Ident(w, t_w)$. According to Lemma 2, there are two

cases, whether in the first case or in the second case, the propagation of the temporal dependencies on $(w, t_w)$ along the edge $\{(v, t_v), (w, t_w)\}$ should be considered $Ident(w, t_w)$ times, hence we have:

$$\delta_{s.}(v, t_v) = \sum_{(v, t_v) \in P_s(w, t_w)} \left( \frac{\sigma_{s(v, t_v)} \cdot Flag(w, t_w)}{\sum_{t' \in T_w} \sigma_{s(w, t')} \cdot Flag(w, t')} \right.$$

$$\left. \cdot Ident(w, t_w) + \frac{\sigma_{s(v, t_v)} \cdot Ident(w, t_w)}{\sigma_{s(w, t_w)}} \cdot \delta_{s.}(w, t_w) \right)$$

The proof completes. □

## C OMITTED ALGORITHMS AND COMPLEXITY ANALYSIS

### C.1 Graph Transformation Algorithm

---
**Algorithm 2:** GTA algorithm

**Input:** original graph $G = (V, E)$
**Output:** time instance graph $G_t = (V_t, E_t)$
1: $V_t \leftarrow \emptyset, E_t \leftarrow \emptyset$
2: **foreach** $(u, v, t_v) \in E$ **do**
3:   $V_t \leftarrow V_t \cup (v, t_v)$
4:   **if** $t <$ the minimum timestamp among those associated with the incoming edges of $u$ or $N_{in}(u) = \emptyset$ **then**
5:    $V_t \leftarrow V_t \cup (u, MIN)$
6: **foreach** $(v, t_v) \in V_t$ and each $v$'s outgoing edge $(v, w, t_w)$ **do**
   // edges are sorted by timestamps
7:   **if** $t_v \leq t_w$ **then**
8:    $E_t \leftarrow E_t \cup \{(v, t_v), (w, t_w)\}$
9: **return** $G_t$

---

The graph transformation algorithm GTA is outlined in Algorithm 2. GTA generates the transformed graph by simply traversing $G$. The time complexity is $O(|E| + \sum_{v \in V} |S(v)| \sum_{w \in N_{out}(v)} |S(w)|)$.

**Size of the time instance graph $G_t$.** The number of vertices in $G_t$ is $|V_t| = \sum_{v \in V} |S(v)|$. The number of edges in $G_t$ depends on the temporal connectivity. The stronger the temporal connectivity, the more the number of edges. If $G$ is weakly temporal connected, $|E_t|$ may be less than $|E|$. The upper bound of $|E_t|$ is $\sum_{v \in V} |E_{in}(v)||E_{out}(v)| + E_{out}(Source)$ if all the paths in $G$ are temporal paths, where $|E_{in}(v)|$ and $|E_{out}(v)|$ are the number of incoming and outgoing edges of $v$, respectively; $E_{out}(Source)$ is the number of outgoing edges of all source vertices.

### C.2 Graph Compression Algorithm

The graph compression algorithm GCA is outlined in Algorithm 3, which compresses the time instance graph by traversing $G_t$ once. The time complexity of GCA is $O(\sum_{v \in V, (v, t_v) \in S(v)} |S(v)|^2 (|N_{in}(v, t_v)| + |N_{out}(v, t_v)|))$, and the space overhead is $O(|V_t| + |E_t|)$.

### C.3 Complexity of OTBC

**Time and space complexities of** OTBC. OTBC takes $G_t'$ as an input, and optimizes ETBC with Lemma 4. But it still needs to compute all-pairs optimal temporal paths, hence the time complexity of OTBC is $O(|V_t'|(|E_t'| + |V_t'|log|V_t'|))$; the space complexity is $O(|E_t'| + |V_t'|)$.

since we need to store matrices of size $|V_t'|$ for the temporal path counts and the temporal dependencies.

---
**Algorithm 3:** GCA algorithm

**Input:** time instance graph $G_t = (V_t, E_t)$
**Output:** compressed graph $G_t' = (V_t', E_t')$
1: **foreach** $(v, t_1)$ and other vertex instances $(v, t_i) \in S(v)$ **do**
   // $S(v)$ is sorted by timestamps
2:   **if** $N_{in}(v, t_1) = N_{in}(v, t_i)$ and $N_{out}(v, t_i) = N_{out}(v, t_i)$ **then**
    // Rule 1
3:    compress $(v, t_1)$ and $(v, t_i)$ to $(v, \{t_1, t_i\})$
4:    compress corresponding adjacent edges
5:   **else if** $N_{in}(v, t_1) = N_{in}(v, t_i)$ **then**
6:    compress edges by Rule 2
7: **return** $G_t'$

---

### C.4 Approximate Algorithm

The pseudocode of ATBC is shown in Algorithm 4. ATBC takes as inputs the compressed *time instance graph* $G_t$, an accuracy parameter $\epsilon$, and a confidential parameter $\delta \in (0, 1)$. It outputs the approximate TBC values $\widehat{TBC}(w)(\forall w \in V)$ that is an $(\epsilon, \delta)$-approximation

---
**Algorithm 4:** ATBC algorithm

**Input:** compressed time instance graph $G_t' = (V_t', E_t')$, accuracy parameter $\epsilon \in (0, 1)$, confidence parameter $\delta \in (0, 1)$
**Output:** $\widehat{TBC}(w)$ for each vertex $w \in V$
1: $|S_0| \leftarrow 0, |S_1| \leftarrow \frac{(1 + 8\epsilon + \sqrt{1 + 16\epsilon})ln(2/\delta)}{4\epsilon^2}, \mathbf{v}_w \leftarrow \{\mathbf{0}\}$
2: $i \leftarrow 1, j \leftarrow 1, \eta \leftarrow \delta/2^i$
3: **while** *True* **do**
4:   **foreach** $j = 1$ *to* $|S_i| - |S_{i-1}|$ **do**
5:    sample $(u, v)$ from $\mathcal{R}$
6:    **foreach** vertex instance $(v, t_v)$ in $S(v)$ **do**
7:     $d(u, (z, t_z)), P_u(z, t_z), \sigma_{u(z, t_z)}, d(u, v), Flag(v, t_v) \leftarrow$ SSSP$(G_t, u, (v, t_v))$ // $(z, t_z)$ is the traversed vertex, $\sigma$ is computed by Eq. 1
8:    $\sigma_{uv} \leftarrow \sum_{(v, t_v) \in S_v, Flag(v, t_v) = 1} \sigma_{u(v, t_v)}$
9:    **foreach** vertex instance $(v, t_v)$ having $Flag(v, t_v) = 1$ **do**
10:     $\forall (z, t_z) \in P_u(v, t_v), \sigma_{(z, t_z)(v, t_v)} \leftarrow 1$
11:     **foreach** $(w, t_w)$ on the shortest path from $u$ to $(v, t_v)$, in reverse order by $d(u, v)$ **do**
12:      $\sigma_{u(v, t_v)}(w, t_w) \leftarrow \sigma_{u(w, t_w)}\sigma_{(w, t_w)(v, t_v)}$; $\sigma_{uv}(w) \leftarrow \sigma_{uv}(w) + \sigma_{u(v, t_v)}(w, t_w)$
13:      **if** all vertex instances $(w, t_w)$ in $S(w)$ is computed **then**
14:       $\mathbf{v}_w \leftarrow \{\mathbf{v}_w \cup \{\frac{\sigma_{uv}(w)}{\sigma_{uv}}\}\}$
15:    $j \leftarrow j + 1$
16:   compute $\omega_i^*, \alpha_i, \bar{ub}_i$ by Eq. 1, 1, 1
17:   **if** $\bar{ub}_i > \epsilon$ **then**
18:    $|S_{i+1}|$ is the minimal positive root of the equation
    $-8((i + 2)ln\frac{2}{\delta})^3 + ((i + 2)ln\frac{2}{\delta})^2(-16\omega_i^* + (1 + 4\epsilon)^2)x - 4((i+2)ln\frac{2}{\delta})(\omega_i^* - \epsilon)^2(1 + 4\epsilon)x^2 + 4(b - f)^4 x^3 = 0;$
19:    $i \leftarrow i + 1$
20:   **else**
21:    **return** $\widehat{TBC}(w) \leftarrow ||\mathbf{v}_w||/|S_i|$ *for each vertex* $w \in V$

---

| $z,t_z$ | $b,\{1,3\}$ | $x,1$ | $c,5$ | $c,7$ | $m,4$ | $y,2$ |
|---|---|---|---|---|---|---|
| $P_a(z,t_z)$ | $a$ | $a$ | $b,\{1,3\}$ | $b,\{1,3\}$ | $b,\{1,3\}$ | $x,1$ |
| $\sigma_a(z,t_z)$ | 2 | 1 | 2 | 2 | 2 | 1 |

| $z,t_z$ | $y,6$ | $d,6$ | $d,\{8,9,10\}$ | $f,11$ |
|---|---|---|---|---|
| $P_a(z,t_z)$ | $c,5$ | $(c,5),(m,4)$ | $(c,5),(c,7)$ | $(d,6),(d,\{8,9,10\})$ |
| $\sigma_a(z,t_z)$ | 2 | 4 | 12 | 16 |

**Figure 4: The values of $P_a(z,t_z)$ and $\sigma_{a(z,t_z)}$**

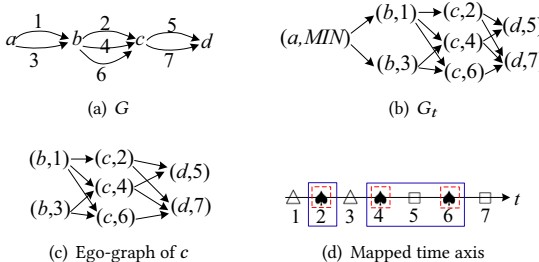

(a) $G$  (b) $G_t$

(c) Ego-graph of $c$  (d) Mapped time axis

**Figure 5: An example of how to compute *vc-ratio*$(v)$ and *ec-ratio*$(v)$**

of the exact values. First, ATBC initializes variables and sample size $|\mathcal{S}_0|$, $|\mathcal{S}_1|$, and samples original vertex pairs (lines 1-5). Then, ATBC computes $\widehat{TBC(w)}$ for each $w \in V$ (lines 6-16). The method used in this step is customized for temporal graphs. Specifically, for each sampled original vertex pair $(u,v)$, ATBC performs the single-source shortest path SSSP algorithm from $u$ to all $v$'s vertex instances $(v,t_v)$. For all the traversed vertex instances $(z,t_z)$, ATBC records the local shortest path distance $d(u,(z,t_z))$, predecessors $P_u(z,t_z)$, the count $\sigma_{u(z,t_z)}$ of local shortest paths, the shortest temporal path distance $d(u,v)$, and the flag $Flag(v,t_v)$ that indicates whether $(v,t_v)$ is the end vertex of an optimal temporal path from $u$ to $v$ (lines 6, 7). Note that, $G_t$ is the compressed *time instance graph*, which includes the equivalent vertices, hence $\sigma_{u(z,t_z)}$ is computed by Eq. 1. If all the vertex instances of $v$ are traversed, ATBC computes $\sigma_{uv}$, starts backtracking $(w,t_w)$ from all vertex instances $(v,t_v)$ of $v$ having $Flag(v,t_v) = 1$ to $u$ along the computed shortest paths, computes $\sigma_{uv}(w) = \sum_{(w,t_w)\in S(w),Flag(v,t_v)=1} \sigma_{u(v,t_v)}(w,t_w)$ and updates vector $\mathbf{v}_w$ (lines 9-16). Thereafter, ATBC checks whether $(\epsilon,\delta)$-approximation is satisfied (i.e., the stopping condition $\bar{ub} \le \epsilon$ holds). If not, ATBC proceeds to sample and iteratively compute $\widehat{TBC(w)}$ until $(\epsilon,\delta)$-approximation is achieved (lines 17-21). Note that, ATBC adopts the same progressive sampling as ABRA[28], the sample size $|\mathcal{S}_i|$ is computed by a heuristic approach. Here we only directly give the sample size (lines 1, 18) and omit the detailed derivation.

**Time and space complexities of** ATBC. ATBC iteratively samples vertex pairs and then performs optimal temporal path computation. In each iteration, optimal temporal paths between the vertex pairs with the same start vertex are all obtained by a single-source shortest path computation. Hence, the time complexity of ATBC is $O(\sum_{i=1}^{l} Count(Aggr(S_i))(|E'_t| + |V'_t||log|V'_t|))$, where $l$ is the number of iterations; $Aggr(S_i)$ denotes the aggregation operation on

$S_i$. The condition of aggregation is that the start vertices of the vertex pairs are the same. After aggregation, we use the Count function to get $Count(Aggr(S_i))$. The space complexity of ATBC is $O(|E'_t| + |V'_t|)$.

## D  EXAMPLES

An example that shows how $\delta_{s.}(v,t_v)$ is computed by Lemma 4 is detailed below.

Example 3. *Consider Figure 2. Take the computation of $\delta_{a.}(z,t_z)$ $(\forall (z,t_z) \in V_t)$ as an example. First traversing the compressed $G_t$ (as plotted in Figure 2(c)) from $a$, we get $P_a(z,t_z)$, $\sigma_{a(z,t_z)}$ and $Flag(z,t_z)$. For any $(z,t_z) \in V_t \setminus \{y,6\}$, $Flag(z,t_z) = 1$. The values of $P_a(z,t_z)$ and $\sigma_{a(z,t_z)}$ are illustrated in Figure 4. For instance,*

$$\sigma_{a(b,\{1,3\})} = 1 \times Ident(b,\{1,3\}) = 2.$$
$$\sigma_{a(d,\{8,9,10\})} = (\sigma_{a(c,5)} + \sigma_{a(c,7)}) \times Ident(d,\{8,9,10\}) = 12.$$

*Next, computing $\delta_{a.}(z,t_z)$, $\delta_{a.}(z)$ by Lemma 4 and Lemma 3, respectively, then we have:*

$$\delta_{a.}(d,6) = \frac{4}{16} = \frac{1}{4}; \delta_{a.}(d,\{8,9,10\}) = \frac{12}{16} = \frac{3}{4}; \delta_{a.}(d) = 1;$$

$$\delta_{a.}(x,1) = \frac{1}{1} = 1; \ \delta_{a.}(x) = 1;$$

$$\delta_{a.}(c,5) = (\frac{2\times3}{16} + \frac{3}{4} \times \frac{2\times3}{12}) + (\frac{2}{16} + \frac{2}{4} \times \frac{4}{16}) = 1;$$

$$\delta_{a.}(c,7) = \underbrace{\frac{2\times3}{16} + \frac{3}{4} \times \frac{2\times3}{12}}_{(d,\{8,9,10\}),(f,11)\text{'s contribution}} = \frac{3}{4}; \ \delta_{a.}(c) = 1\frac{3}{4};$$

$$\delta_{a.}(m,4) = \underbrace{\frac{2}{16} + \frac{2}{4} \times \frac{1}{4}}_{(d,6),(f,11)\text{'s contribution}} = \frac{1}{4}; \ \delta_{a.}(m) = \frac{1}{4};$$

$$\delta_{a.}(b,\{1,3\}) = (\frac{2}{4} + 1\times\frac{2}{2}) + (\frac{2}{4} + \frac{3}{4}\times\frac{2}{2}) + (\frac{2}{2} + \frac{1}{4}\times\frac{2}{2}) = 4.$$

## E  DISCUSSIONS

### E.1  Compression Ratio Estimation

Compression ratios include vertex compression ratio *vc-ratio* and edge compression ratio *ec-ratio*, which are defined as *vc-ratio* $= 1 - \frac{|V'_t|}{|V_t|}$ and *ec-ratio* $= 1 - \frac{|E'_t|}{|E_t|}$. Intuitively, graph temporal connectivity and distribution of timestamps have effects on the compression ratios, while it is difficult to give a specific formula for the compression ratios. So we define the compression ratios of vertex $v$ and $v$'s adjacent edges as *vc-ratio*$(v)$ and *ec-ratio*$(v)$, respectively, and use *vc-ratio*$(v)$ (resp. *ec-ratio*$(v)$) to estimate *vc-ratio* (resp. *ec-ratio*). Considering that vertex or edge compression depends on the sets of neighbors, we define an ego-graph of vertex $v$ as a subgraph of $G_t$ induced by the vertex set $\cup_{\forall(v,t_v)\in S(v)} N(v,t_v) \cup S(v)$. We map the vertex instances in the ego-graph of $v$ into a time axis according to the timestamps. For convenience of explanation, in the time axis, let ♠, △, □ denote $v$'s instances, in-neighbors, and out-neighbors, respectively. All the $v$'s instances consist of ♠ space. Based on this, *vc-ratio*$(v)$ and *ec-ratio*$(v)$ are easily computed, as stated below.

*vc-ratio*$(v)$: Let $I_1, I_2, \cdots, I_n$ denote the ♠ subspaces divided by △ and □. Take Figure 5 as an example. Figure 5(a) and Figure 5(b) depict the original temporal graph $G$ and the time instance graph $G_t$, respectively. Figure 5(c) illustrates the ego-graph of $c$. Figure 5(d)

plots the mapped time axis. $\triangle$ and $\square$ divide the $\spadesuit$ space into 3 subspaces, each of which is surrounded by a red dashed rectangle. According to the vertex compression rule, the vertex instances in the same subspaces are compressed into an equivalent vertex, then $vc\text{-}ratio(v) = 1 - \frac{n}{|S(v)|}$.

$ec\text{-}ratio(v)$: There are two cases involving edge compression. (i) When equivalent vertex instances $(v, t)$ are compressed into a vertex instance (above-mentioned vertex compression), the corresponding edges are compressed. The reduced number of edges is $N_1 = \sum_{i=1}^{n}(Count(I_i) - 1)|N(v, t_v)|$, where $Count(I_i)$ is the number of $v$'s instances in subspace $I_i$. (ii) When the in-neighbors of $v$'s instances are the same, we do edge compression. Let $I_1', I_2', \cdots,$ $I_m'$ denote the $\spadesuit$ subspaces divided by $\triangle$. According to the edge compression rule, $N_2 = \sum_{i=1}^{m}(Count(I_i') - 1)$ virtual edges are created, $N_3 = \sum_{i=1}^{m}(Min_{(v, t_v) \in I_i'}|N(v, t_v)|(Count(I_i') - 1))$ edges are reduced at most. Then $ec\text{-}ratio(v) \approx \frac{N_1 - N_2 + N_3}{\sum_{(v, t_v) \in S(v)}|N(v, t_v)|}$.

We could uniform randomly sample a subset $S$ of vertices, and use $\frac{\sum_{v \in S} vc\text{-}ratio(v)}{|S|}$ and $\frac{\sum_{v \in S} ec\text{-}ratio(v)}{|S|}$ as $vc\text{-}ratio$ and $vc\text{-}ratio$ estimation (denoted as $\widehat{vc\text{-}ratio}$ and $\widehat{ec\text{-}ratio}$), respectively.

### E.2 Support for Other Paths

(i) Calculating TBC based on ETP is similar to that based on STP. The difference is that in the first phase of ETBC or OTBC, calculating TBC based on ETP maintains the earliest end time instead of the shortest temporal path distance.

(ii) The optimization techniques are suitable for weighted temporal graphs as long as a new condition (edge weights are equal) is added to the vertex instance and edge compression.

(iii) The proposed techniques are also applicable to other types of optimal temporal paths, such as the fastest temporal path (FTP), which is the path with the minimum duration (i.e., minimum $E_p - S_p$). Calculating TBC based on FTP is equivalent to that based on STP over the weighted *time instance graph*. The edge weight from vertex instance $(u, t_u)$ to $(v, t_v)$ is the time duration $t_v - t_u$.

(iv) The combination of STP and ETP, i.e., shortest earliest betweenness, is directly supported by the proposed method.

(v) For the latest departure path (LDP), the proposed method needs to do the following adaption to support it. (a) In Eq. 1, otherwise $s = v$, $\sigma_{s(w, t_w)} = \sigma'_{s(w, t_w)} \cdot Ident(w, t_w)$ changes to $\sigma_{s(w, t_w)} = \sigma'_{s(w, t_w)}$. (b) A vertex instance $(w, t_w)$ needs to record and constantly update the predecessors and corresponding latest departure time until it finds the final latest departure time and predecessors.

In summary, if the local optimal temporal path (i.e., optimal temporal path between the time instances) satisfies the subpath optimality property in the time instance graph, then it could be supported by the proposed method; but the proposed method may need to make a little adaption work by the characteristics of different paths to support iterative TBC computation.

### E.3 Dynamic Update

Considering that vertex insertion and deletions are often accompanied by edge insertion and deletions, we discuss the insertion and deletions of a new temporal edge $e = (u, v, t_v)$.

**Edge insertion**. If $(v, t_v)$ is not in the time instance graph $G_t$, then we create $(v, t_v)$. If $u$ is a new vertex or the timestamps associated with the incoming edges of $u$ are all greater than $t$, then we create $(u, MIN)$ in $G_t$. $\forall (u, t_u) \in S(u)$, if $t_u \le t_v$, then we insert an edge from $(u, t_u)$ to $(v, t_v)$. $\forall w \in N_{out}(v)$, $(w, t_w) \in S(w)$, if $t_v \le t_w$, then we insert an edge from $(v, t_v)$ to $(w, t_w)$. All the time instances of $u, v$ and $w$ are the affected vertices, we compress these vertices and their adjacent edges by the compression rules 1 and 2, and then the updated compressed time instance graph is obtained.

**Edge deletion**. The process of edge deletion is similar to that of edge insertion, except that we delete edges rather than insert edges.

**TBC update**. Let $R_{in}(v)$ be the set of vertices that can reach $v$, and $R_{out}(u)$ be the set of vertices that $u$ can reach. $R_{in}(v)$ and $R_{out}(u)$ can be obtained by a 2-hop labeling index. Then the potential affected vertices are those lying on the paths from any vertex in $R_{in}(v)$ to the ones in $R_{out}(u)$. We update TBC values of the affected vertices $y$ by recomputing $\delta_{x.}(y)$ ($\forall x \in R_{in}(v)$).

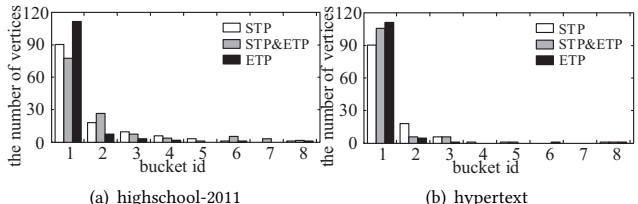

(a) highschool-2011      (b) hypertext

**Figure 6: TBC distribution histogram**

**Table 6: Statistics of the real and estimated compression ratios**

| Datasets | Compression | | Estimation | |
|---|---|---|---|---|
| | *vc-ratio* | *ec-ratio* | $\widehat{vc\text{-}ratio}$ | $\widehat{ec\text{-}ratio}$ |
| highschool-2011 | 0.686 | 0.878 | 0.716 | 0.867 |
| highschool-2012 | 0.686 | 0.925 | 0.603 | 0.890 |
| highschool-2013 | 0.620 | 0.887 | 0.609 | 0.886 |
| hypertext | 0.514 | 0.763 | 0.481 | 0.765 |
| hospital-ward | 0.493 | 0.819 | 0.465 | 0.845 |
| primaryschool | 0.294 | 0.620 | 0.234 | 0.608 |
| infectious | 0.385 | 0.579 | 0.411 | 0.564 |
| emails | 0.115 | 0.253 | 0.116 | 0.230 |
| wikivoyage-it | 0.770 | 0.972 | 0.791 | 0.955 |
| wikiedits-se | 0.493 | 0.708 | 0.521 | 0.651 |
| mathoverflow | 0.184 | 0.412 | 0.228 | 0.394 |
| superuser | 0.119 | 0.383 | 0.142 | 0.404 |
| wikitalk | 0.224 | 0.390 | 0.266 | 0.427 |

## F ADDITIONAL RESULTS

### F.1 TBC Distribution

We distribute the vertices to 8 evenly distributed buckets between 0 and the highest temporal betweenness value. Figure 6 shows the experimental results on datasets highschool-2011 and hypertext, others are omitted due to limited space. It is observed that, no matter what types of optimal temporal paths, the TBC values still follow a power-law like distribution.

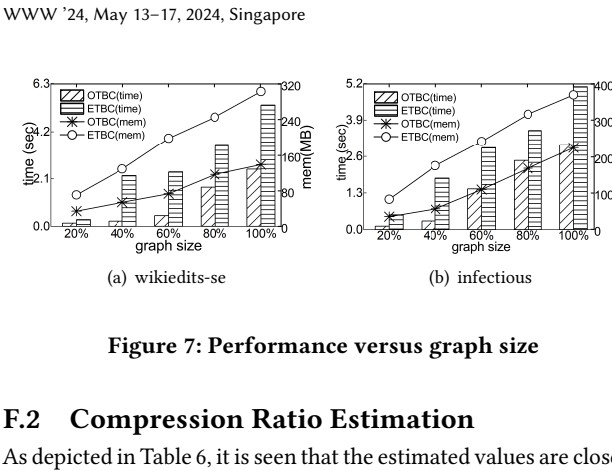

(a) wikiedits-se

(b) infectious

**Figure 7: Performance versus graph size**

## F.2 Compression Ratio Estimation

As depicted in Table 6, it is seen that the estimated values are close to the true values. For example, on hypertext, $\widehat{vc\text{-}ratio}$ = 0.481 ($vc\text{-}ratio$ = 0.514) and $\widehat{ec\text{-}ratio}$ = 0.765 ($ec\text{-}ratio$ = 0.763); on emails, $\widehat{vc\text{-}ratio}$ = 0.116 ($vc\text{-}ratio$ = 0.115) and $\widehat{ec\text{-}ratio}$ = 0.230 ($ec\text{-}ratio$ = 0.253). These show the effectiveness of the compression strategies and the estimation method proposed in Section 4.3.

## F.3 Scalability

We vary the size of the temporal graph and report the performance of ETBC and OTBC on wikiedits-se and infectious in Fig 7. It is seen that the computation costs and memory consumption grow with the increase in the graph size. This is consistent with the complexity analyses in Section 4.

## F.4 Performance of Dynamic Update

We evaluate the insertion (resp. deletion) performance when changing the number of inserted (resp. deleted) edges from 10K to 50K in 10K increments. Figure 8 shows the results. As observed, the update cost ascends with the number of inserted or deleted edges increases because of the growing size of the affected vertex pairs, and it is up to 2 times faster than the re-computation cost.

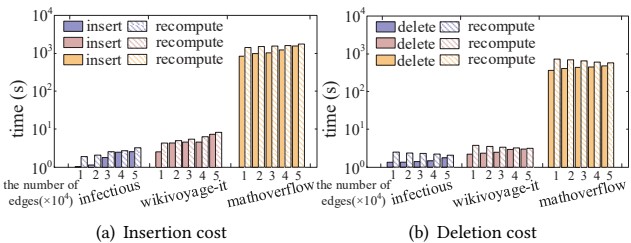

(a) Insertion cost

(b) Deletion cost

**Figure 8: Performance versus TBC update**

Received 12 October 2023; revised xx xx 2023; accepted xx xx 2023

