# OpenReview forum: "Efficient Exact and Approximate Betweenness Centrality Computation for Temporal Graphs"
_ACM.org/TheWebConf/2024/Conference — TheWebConf24 Oral_

### Official Review · Reviewer_MivX · 2023-11-24

**Novelty:** 5
**Technical Quality:** 6

**Review:**

The paper proposed a novel betweenness centrality extracting method for temporal graph, there are three parts of contribution, a temporal influence accumulating method ETBC, a compression method OTBC to accelerate TBC computation, and an approximate method ATBC to further reduce the algorithm complexity. The proposed method is technical sound and prove to be effiecient in both theoretical analysis and experiment performance.

**Questions:**

There are some questions that may help me understand more about the proposed method:
1. Can the proposed graph instance extracting and compressing method helps to compute other graph statistical features like cluster coefficiency and graph entropy?
2. Can the proposed temporal influence accumulating method support temporal Resistance Distance computing?
3. What is the detailed distribution of betweenness centrality for each node in the temporal graph under different extracting methods?
4. For network updation, how does the error rate accumulate by the edge insertion or deletion? How does the initial graph size influence such accumulating?

**Reviewer Confidence:**

3: The reviewer is confident but not certain that the evaluation is correct

**Scope:**

4: The work is relevant to the Web and to the track, and is of broad interest to the community

---

### Official Review · Reviewer_6UGA · 2023-11-24

**Novelty:** 5
**Technical Quality:** 7

**Review:**

The topic and paper is very interesting. The authors investigate on the problem of computing betweenness centrality in temporal graphs (in this case, on discrete time). The  authors provide and exact and approximate algorithm, and run it on real-world datasets.

My only question, that could be addressed during the rebuttal, is regarding the related works; the authors do not consider fastest paths, even though they make sense in the setting of betweenness centrality (more than foremost for example, which induce a strong bias on time). The authors missed lines of work that already define and compute BW centrality on temporal graphs. See the following papers :

Stream graphs and link streams for the modeling of interactions over time
Temporal betweenness centrality in dynamic graphs
Temporal node centrality in complex networks
Temporal Betweenness Centrality on Shortest Paths Variants

The authors addressed my comments in a satisfactory manner.

**Questions:**

See above

**Ethics Review Description:**

--

**Reviewer Confidence:**

4: The reviewer is certain that the evaluation is correct and very familiar with the relevant literature

**Scope:**

4: The work is relevant to the Web and to the track, and is of broad interest to the community

---

### Official Review · Reviewer_zWci · 2023-11-27

**Novelty:** 7
**Technical Quality:** 7

**Review:**

The paper proposes new algorithms to compute Temporal Betweenness Centrality (TBC), along with theoretical guarantees and empirical results.

The first contribution is the definition of a transformed time instance graph and the derivation of a new recursive temporal dependency formulation, which is at the core of the efficiency of the proposed algorithms.

The first algorithm proposed, called Exact Temporal Betweenness Centrality (ETBC), arises directly from the transformed time instance graph and the new recursive temporal dependency formulation, and it is suitable for different optimal temporal paths (namely, shortest temporal path, earliest temporal path, and their combination).

The second algorithm proposed, called Optimized Temporal Betweenness Centrality (OTBC), is, as the name suggests, an optimized (and still exact) version of ETBC. The optimization is obtained by compressing vertices and edges of the time instance graph. As a matter of fact, I see no advantage in using ETBC instead of OTBC, but having both in the paper is helpful in understanding the derivation process that ultimately led to OTBC.

Finally, an approximate algorithm, called Approximate Temporal Betweenness Centrality (ATBC), is introduced. Error upper bounds are derived using Rademacher Averages.


Pros:
- A new recursive temporal dependency formulation is introduced.
- Both the exact and approximate algorithms proposed are really efficient.
- Extensive experiments are performed, all methods compared are implemented in C++ and the entire experimental setup seems really fair.


Cons:
- Perhaps a more discursive style would help the reader better understand all the ideas behind the article. On the other hand I understand that there is a limit of 8 pages and in these cases a more concise and formal form is preferred.

---

I have read other reviewers questions and authors responses and I confirm my evaluation as it is.

**Questions:**

The following is a mix of questions and suggestions:
- The quality of Figure 1 can be improved. Additionally, adding at least two parallel edges would help the reader better interpret the meaning of the illustrated toy example.
- (Lines 310-314) When you say "the subpaths of optimal temporal paths may not be optimal" it may be useful to provide a brief counterexample for each of the two optimal temporal paths considered.
- In tables, especially Table 4 which extensively uses numbers with three decimal places, it is hard to distinguish between commas and dots. Maybe consider using spacing instead of commas to represent large numbers.
- (Lines 868-869) "OTBC does not finish on wikitalk". In Table 4 no error is reported for wikitalk (as expected). But column "OTBC" contains values, is it a typo?

**Ethics Review Description:**

None.

**Reviewer Confidence:**

3: The reviewer is confident but not certain that the evaluation is correct

**Scope:**

4: The work is relevant to the Web and to the track, and is of broad interest to the community

---

### Official Review · Reviewer_ijXx · 2023-11-27

**Novelty:** 3
**Technical Quality:** 1

**Review:**

Summary: the goal of this paper is to introduce a new way of finding the betweenness centrality in temporal graphs.
In static graphs, shortest paths are used in the definition of betweenness centrality. In temporal graphs, any optimal path can replace the role of shortest paths in the definition: earlier finish time path, fastest path, shortest path etc.
One main problem in design of efficient algorithms for betweenness centrality in temporal graphs is that the recursive formula of Brandes [1] is invalid in temporal graphs.

The papers proposes transforming a given temporal graph to a static graph which they call time instance graph (definition presented in lines 320-342). In lemma 2 (line 367) they show a recursive formula which is similar to the recursive formula of Brandes. Using this formulation they claim that they can design an efficient algorithm for all the definitions of betweenness centrality in temporal graphs.

[1] Brandes, A faster algorithm for betweenness centrality 2001, Journal of mathematical sociology.

Cons: (1) the paper is hard to read and there are many mistakes in the paper. (2) I am not sure if the results are valid.


Detailed comments:

Line 344: in the definition of \delta_{s\cdot } the sums should be over all v. The way it is written, it seems like it is over the triple of s,z,v . Same notation problem in definition 6 .

In definition 6 is s a vertex in the original graph or a vertex instance in time instance graph?

In algorithm 1. How can the FLAG be obtained from a shortest path algorithm so that it is adaptable to earlier/shortest and all other concepts of optimality?

I didn't understand the proof of lemma 2. In particular what is \sigma_{sz}((v,t_v), {(v,t_v), ((w,t_w))})


----

After reading the authors explanation, I still don't have an answer to my questions.

I don't have any intuition why the results would hold true, and I got lost verifying some proofs. I think the paper would benefit from major revisions.

**Questions:**

In definition 6 is s a vertex in the original graph or a vertex instance in time instance graph?

In algorithm 1. How can the FLAG be obtained from a shortest path algorithm so that it is adaptable to earlier/shortest and all other concepts of optimality?

I didn't understand the proof of lemma 2. In particular what is \sigma_{sz}((v,t_v), {(v,t_v), ((w,t_w))})

**Ethics Review Description:**

no issue

**Reviewer Confidence:**

2: The reviewer is willing to defend the evaluation, but it is likely that the reviewer did not understand parts of the paper

**Scope:**

3: The work is somewhat relevant to the Web and to the track, and is of narrow interest to a sub-community

---

### Decision · Program_Chairs · 2024-01-22

**Decision:**

Accept (Oral)

**Comment:**

This paper considers the computation of Temporal Betweenness Centrality in temporal graphs, providing algorithms with theoretical guarantees along with experimental results.
 The paper is quite technical and hard to read, but the results are interesting and valuable, and I could not find flaws, neither did any of the reviewers as far as I can see.
 Especially one of the reviewers was very critical due essentially to some missing intuitions that could help the reader understand the definitions and why/how they are sound.
 The same problem was also raised by another reviewer, that was anyway finally convinced of the soundness of the results.
 The first reviewer, even after a long debate with the authors, concluded that (s)he was still unsatisfied.
 I had a read myself and I agree on the fact that the technical part does not help much the reader.
 Nonetheless, I think that the authors did a good job in presenting their proofs in the appendix, and did their best during the discussion to convince the reviewers.
 They at least convinced me. My suggestion is that, in case of acceptance, they try to follow the directions suggested by the reviewers to improve the writing as much as possible.